# Effectiveness of acupuncture therapy for the prevention of emergence agitation in children: A systematic review and meta-analysis with trial sequential analysis

Takahiro Mihara[1,2,☯]*, Daisuke Nakajima[1,3☯], Toshiyuki Hijikata[1], Makoto Tomita[1], Takahisa Goto[3]

1 Department of Health Data Science, Yokohama City University Graduate School of Data Science, Yokohama, Japan, 2 Department of Anesthesiology, Yokohama City University, School of Medicine, Yokohama, Japan, 3 Department of Anesthesiology, Yokohama City University Medical Center, Yokohama, Japan

☯ These authors contributed equally to this work.
* meta.analysis.r@gmail.com

**Data Availability Statement:** All relevant data are within the paper and its Supporting information files.

## Abstract

This study aimed to evaluate the effectiveness of acupuncture therapy in preventing emergence agitation (EA) in children. A systematic review and meta-analysis were conducted across multiple locations according to the articles searched. Seven databases, including trial registration sites, were searched. A total of six trials were included involving 489 patients; of them, 244 received acupuncture therapy. Randomized clinical trials (RCTs) evaluating the incidence of EA compared with placebo/sham or standard care in children were included. The primary outcome was the incidence of EA, as evaluated using a specific assessment tool. Data about the incidence rate of EA, heterogeneity, quality of trials and evidence, and adverse events were collected. Additionally, data about patient demographic characteristics, type of anesthesia, duration and onset of acupuncture therapy, EA and pain score, time taken for extubation, and post-anesthesia care unit length of stay were collected. The results indicated that the overall incidence of EA in the acupuncture therapy group and the control group was 23.4% and 39.5%, respectively, with no significant difference (risk ratio, 0.62; 95% confidence interval, 0.26–1.48; $I^2$ = 63%). Subgroup analysis showed a significant difference in the overall incidence of EA in the acupuncture therapy and control groups according to surgery type (high-risk vs. low-risk surgery), suggesting that acupuncture therapy may be effective in reducing EA for patients undergoing high-risk surgery. The quality of evidence was downgraded to "very low" due to the study designs, inconsistency, and possible publication bias. In conclusion, this meta-analysis shows that the currently available RCTs are insufficient to determine the effectiveness of acupuncture therapy in preventing EA in children undergoing general anesthesia.

**Funding:** The author(s) received no specific funding for this work.

**Competing interests:** The authors have declared that no competing interests exist.

## Introduction

Emergence agitation (EA) is a significant concern among pediatric patients undergoing general anesthesia, with a reported incidence ranging from 10% to 80% [1]. Sikich and Lerman [2] characterized emergence delirium, a primary factor contributing to EA, as follows: "a disturbance in a child's awareness of and attention to his or her environment with disorientation and perceptual alterations including hypersensitivity to stimuli and hyperactive motor behavior in the immediate post-anesthesia period." Restless recovery from anesthesia may increase the risk of self-injury and require additional administration of sedatives or analgesic medications, which may delay the patient's discharge from the hospital. Moreover, EA often requires constant nursing supervision, which strains the nursing staff and induces stress in caregivers and families [1, 3].

Pharmacological agents are commonly administered to prevent EA, including propofol, fentanyl, dexmedetomidine, and clonidine [4–8]; however, these pharmacological agents may often cause adverse events, such as delayed recovery, vomiting, and respiratory depression [8, 9]. Therefore, alternative methods to prevent EA are necessary. One suggested method is acupuncture therapy, which has been shown to prevent EA with no adverse event and no requirement for pharmacotherapy [10, 11]; however, its efficacy remains controversial [12].

In children, acupuncture therapy has been found to be a safe and promising complementary or alternative treatment method because of its efficacy and low risk [13]. Stimulating an acupuncture point on the wrist (P6 acupoint stimulation) during the perioperative period can reduce the risk of postoperative nausea and vomiting in children [14]. A potential acupoint for suppressing emergence agitation in pediatric anesthesia is HT7 (Shenmen). HT7 is recognized as an acupoint that promotes tranquility [15]. In animal studies, the therapeutic effects of HT7 acupuncture therapy on psychiatric and neurological disorders have been documented [16, 17]. However, the effectiveness of acupuncture therapy for preventing EA in children remains unclear [12]; additionally, meta-analyses regarding its effectiveness remain lacking. Furthermore, the methodologies of these previous studies varied according to acupuncture therapy, stimulation point selection, and surgery type.

The primary purpose of this meta-analysis was to evaluate the effectiveness of acupuncture therapy in preventing EA in children undergoing general anesthesia. The secondary purpose was to assess adverse events and conduct subgroup analyses based on the following predefined factors: (i) method of acupuncture therapy; (ii) selection of points (unilateral or bilateral); and (iii) type of surgery (high-risk surgery for EA or not).

## Materials and methods

We conducted a systematic review with meta-analysis and trial sequential analysis (TSA). This meta-analysis was performed according to the recommendations of the Preferred Reporting Items for Systematic Reviews and Meta-Analyses (PRISMA) statement [18] and the Cochrane Handbook [19]. Our study protocol and methods were pre-specified and registered on the University Hospital Medical Information Network (Registration number: 000040775) and has been published previously elsewhere [20].

### Search strategy

We searched PubMed, Cochrane Central Register of Controlled Trials, Embase, and Web of Science databases. We also searched the databases of clinicaltrials.gov, the European Union Clinical Trials Register, the World Health Organization International Clinical Trials Registry Platform, and the Japanese University hospital Medical Information Network Clinical Trials

Registry. We also searched the reference lists of the retrieved articles. The last search was conducted on April 20, 2022. The PubMed search strategy is provided in S1 Appendix in S1 File.

To exclude irrelevant articles, two authors (D.N. and T.H.) independently assessed the suitability of the titles and abstracts of the studies identified by the search strategies. We retrieved the full-text versions of potentially relevant studies selected by at least one author, and those that met the inclusion criteria were examined separately. Discrepancies were resolved by consensus through discussion between the two authors.

## Inclusion criteria

We include randomized clinical trials (RCTs) that evaluated the effectiveness of acupuncture therapy in preventing EA compared with a placebo, no medication, or standard care in children undergoing general anesthesia. Articles with the following characteristics were excluded from the study: (i) the incidence of EA was not evaluated using a specific assessment tool such as the Pediatric Anesthesia Emergence Delirium (PAED) scale, Aono's scale, and other numeric scales; (ii) participants were aged >18 years; and (iii) the articles were not based on a study following the RCT design, such as case reports, observational studies, comments, reviews, and animal studies. Eligibility was not restricted by language, type of surgery, or the anesthetic technique.

## Outcomes

**Primary outcome.** The primary outcome was the incidence of EA evaluated using a specific assessment tool. EA incidence was defined according to the criteria established in each study. In cases where EA was classified by severity, the numbers of patients were identified for all degrees of severity and summated. In cases where EA was evaluated at several time points, the time point immediately after emergence (i.e., the earliest time point in the post-anesthesia care unit (PACU) or recovery room) was used to extract the data representing acute EA.

**Secondary outcomes.** The secondary outcomes were the absolute values of EA and pain scores evaluated using specific assessment tools. Additionally, the incidence of adverse events such as nausea and vomiting, time taken for extubation, and PACU length of stay were analyzed.

## Data collection

A standardized, pre-piloted form was used to extract data from the included studies. The extracted information included: (i) number of patients in the study; (ii) age; (iii) sex; (iv) classification of physical status according to the American Society of Anesthesiologists; (v) risk factors for EA; (vi) type of anesthesia; (vii) type of surgery; (viii) method of acupuncture therapy; (ix) duration and onset of acupuncture therapy; (x) number of patients with EA; (xi) absolute values of the EA score evaluated using a specific assessment tool; (xii) absolute values of the pain score evaluated using a specific assessment tool; (xiii) time taken for extubation; (xiv) PACU length of stay; and (xv) adverse effects of acupuncture therapy. Two review authors extracted the data independently, and discrepancies were resolved through discussion. Missing data were requested from the study authors. The study was excluded if the trial authors did not have the missing data.

## Assessment of the risk of bias

We assessed the risk of bias using the Cochrane Risk of Bias tool 2.0 [21] for RCTs, which has five domains and one overall risk of bias domain as follows: (i) bias arising from the

randomization process; (ii) bias due to deviations from intended interventions; (iii) bias due to missing outcome data; (iv) bias in the measurement of outcomes; (v) bias for selection of the reported result; and (vi) overall risk of bias. The risk of bias was assessed as "low," "of some concern," or "high" in each domain.

## Data synthesis and statistical analyses

Statistical analyses were performed using the R statistical software package, version 4.0.3 (R Foundation for Statistical Computing, Vienna, Austria). Dichotomous data were compared between the groups using the risk ratio and continuous data using the mean difference (MD). The risk ratio and MD were summarized using a 95% confidence interval (CI); statistical significance was set at a 95% CI value of 0 or 1 for continuous or dichotomous data, respectively. A random-effect model (DerSimonian and Laird method [22]) was used to combine the results. In addition, we used the Hartung-Knapp-Sidik-Jonkman adjustment method [23] if the number of studies to be combined was small (i.e., $< 10$). Heterogeneity was quantified using the $I^2$ statistic, which was significant if the value exceeded 50%. Subgroup analyses were conducted to explore possible causes in cases with high heterogeneity. A forest plot was used to represent the effect of treatment graphically. A small study effect was assessed using a funnel plot and Egger's regression asymmetry test [24], which was positive if the latter test showed $p < 0.1$.

Subgroup analyses were conducted according to the following predefined factors when the $I^2$ statistic exceeded 50%: (i) method of acupuncture therapy, (ii) selection of points (unilateral or bilateral), and (iii) type of surgery (high-risk vs. low-risk surgery). High-risk surgery for EA was defined as tonsillectomy or adenotonsillectomy. Sensitivity analysis, which was restricted to the studies with a low risk of bias, was performed for the primary outcome.

## Trial sequential analysis

For the primary outcome, TSA was performed to correct for random error and repetitive testing of accumulating and sparse data. TSA monitoring boundaries (i.e., monitoring boundaries for meta-analysis) and required information size (RIS) were quantified, and adjusted CIs were calculated. The risk of type 1 error was maintained at 5% with a power of 90%. A reduction in the risk ratio of 25% was considered clinically significant. If the cumulative Z-curve could not cross the TSA monitoring boundaries, the quality of evidence was downgraded due to inaccuracies in the results.

## Summary of evidence

The Grading of Recommendations Assessment, Development, and Evaluation (GRADE) approach was used to evaluate the quality of evidence of the primary outcomes [25, 26]. Judgments of the quality of evidence were based on the presence or absence of the following variables: (i) limitations in study design; (ii) inconsistency; (iii) indirectness; (iv) inaccuracies in the results; and (v) publication bias. The quality of evidence for the primary outcomes was graded as "very low," "low," "moderate," or "high." Additionally, GRADEpro GDT (https://gradepro.org/) was used to summarize the findings using a table.

## Results

### Search selection and study characteristics

Initially, the database search identified 2,251 articles; among them, the full text of 69 articles was examined in detail. Finally, six trials were included with 489 patients; of them, 244

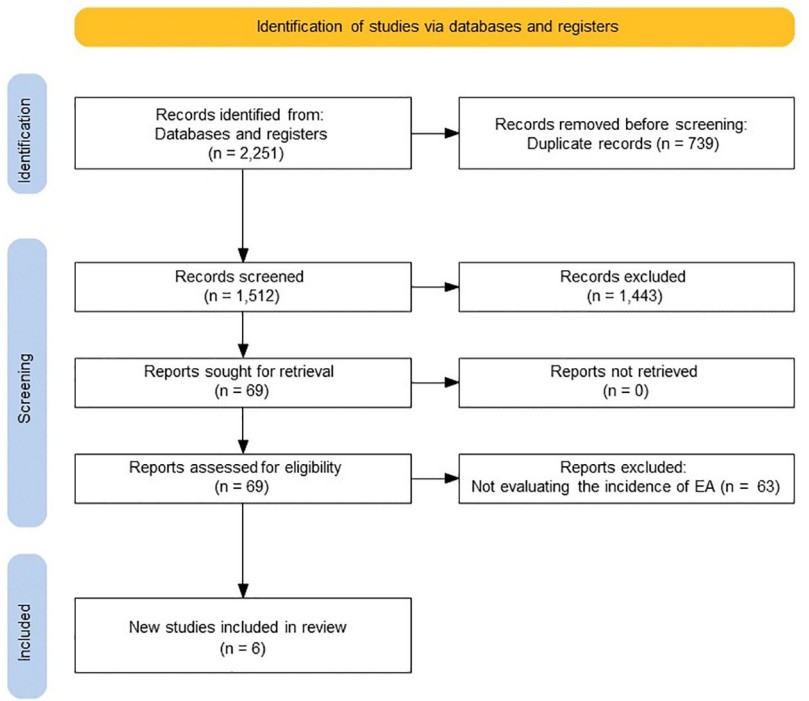

**Fig 1. PRISMA flow diagram.**

received acupuncture therapy (Fig 1). Five trials evaluated the incidence of EA using the PAED scale [3, 10, 12, 27], and one trial evaluated the incidence of EA using a four-point scale [11]. Martin et al. [27] divided the patients into four groups by acupuncture therapy and midazolam premedication; therefore, we analyzed the data separately as two trials: "Martin 2020 MDZ+" and "Martin 2020 MDZ-".

The characteristics of the six trials are presented in Table 1. All trials compared acupuncture therapy with a sham procedure or standard care. Two trials used a peripheral nerve stimulation device for acupuncture therapy [3, 12], three trials used acupuncture needles [11, 27, 28], and one trial used capsicum plasters [10]. All trials stimulated HT7. Furthermore, two trials stimulated LI-4 [11] or the ear [27] and one trial stimulated P6, KI6, and the body [28]. Patient age ranged from 1 to 12 years. Five trials used sevoflurane to maintain general anesthesia, and one trial used isoflurane [28]. When data about our meta-analysis outcomes could not be extracted from the trial's published data, we contacted the corresponding authors for clarification. However, we were unable to receive data from one trial's authors [11] and were consequently unable to estimate the incidence of EA in that trial.

## Risk of bias in included trials

The risk of bias for the primary outcome in the included trials is summarized in Fig 2. We found that three trials had a low risk of bias for our primary outcome, while two had a high risk of bias. The risk of bias for secondary outcomes are summarized in S6 Fig in S1 File.

## Intervention effects for EA

Five trials evaluated the incidence of EA using the PAED scale [3, 10, 12, 27, 28]. The overall incidence of EA in the acupuncture therapy group and the control group was 39.5% and

**Table 1. Characteristics of the trials.**

| Source | Study/ Control | Age (protocol) | Type of surgery | Method of acupuncture | Type of control | Point of acupuncture | laterality | Duration and the onset of acupuncture therapy | Diagnosis tool for EA | Diagnostic criteria for EA |
|---|---|---|---|---|---|---|---|---|---|---|
| Acar 2012 | 25/25 | 2–10 | Elective adenoidectomy and/ or tonsillectomy | Capsicum plasters (5.5 cm2) | Inactive plasters (5.5 cm2) | HT7 | Bilateral | After the induction of anesthesia— throughout the surgery | PAED score | A score of ≥10 at any measurement time was accepted as |
| Hijikata 2016 | 60/60 | 18 m–96 m | Inguinal hernia repair, adenoidectomy and/ or tonsillectomy, unilateral strabismus surgery, cryptorchidism repair, tympanostomy tube insertion and other minor procedures | Single-twitch stimulation at 1 Hz | Electrodes alone were attached; an electrical stimulus was not applied | HT7 | Bilateral | Throughout the operation | PAED score, Aono's score | PAED ≥10 |
| Ismail 2021 | 30/30 | 3–12 y | Elective adenotonsillectomy | Sterile single-use acupuncture needles penetrated the skin to a depth of 1 cm | Sham acpuncture | Ht7, li4, ki6, p6, ear, body | Unilateral | After induction of anesthesia— until the end of surgery | PAED score | PAED ≥10 |
| Lin 2009 | 30/30 | 1–6 y | Bilateral myringotomy and tympanostomy tube placement | A stainless-steel acupuncture needle, 30 mm in length and 0.18 mm in diameter | No acupuncture treatment | Ht7, li-4 | Bilateral | Each acupuncture needle was manually manipulated for 10 s and kept in situ for a total duration of 10 min. | Emergence agitation was assessed on a 4-point scale: | 1 = asleep, calm; 2 = mildly agitated but easily consolable; 3 = moderately agitated or restless and inconsolable; and 4 = hysterical, crying inconsolably, or thrashing |
| Martin 2020 MDZ + | 25/25 | 1–6 y | Unilateral or bilateral myringotomy tube placement only | Seirin pionex press needles (1.8 mm long with an adhesive backing) | No acupuncture treatment | Ht7, ear shen men | Bilateral | After anesthesia induction and removed after completion of the myringotomy tube placement, but prior to leaving the operating room | PAED score | None |

*(Continued)*

**Table 1.** (Continued)

| Source | Study/ Control | Age (protocol) | Type of surgery | Method of acupuncture | Type of control | Point of acupuncture | laterality | Duration and the onset of acupuncture therapy | Diagnosis tool for EA | Diagnostic criteria for EA |
|---|---|---|---|---|---|---|---|---|---|---|
| Martin 2020 MDZ- | 24/25 | 1–6 y | Unilateral or bilateral myringotomy tube placement only | Seirin pionex press needles (length of 1.8 mm with an adhesive backing) | No acupuncture treatment | Ht7, ear shen men | Bilateral | After anesthesia induction and removed after completion of the myringotomy tube placement, but prior to leaving the operating room | PAED score | None |
| Nakamura 2018 | 50/50 | 18 m–96 m | Inguinal hernia, other minor surgery | Single-twitch stimulation at 1 Hz | Electrodes alone were attached; an electrical stimulus was not applied | Ht7 | Unilateral | during the surgery | PAED score, Aono's score | EA: PAED ≥10 |

Abbreviations: EA, emergence agitation; PAED, Pediatric Anesthesia Emergence Delirium scale

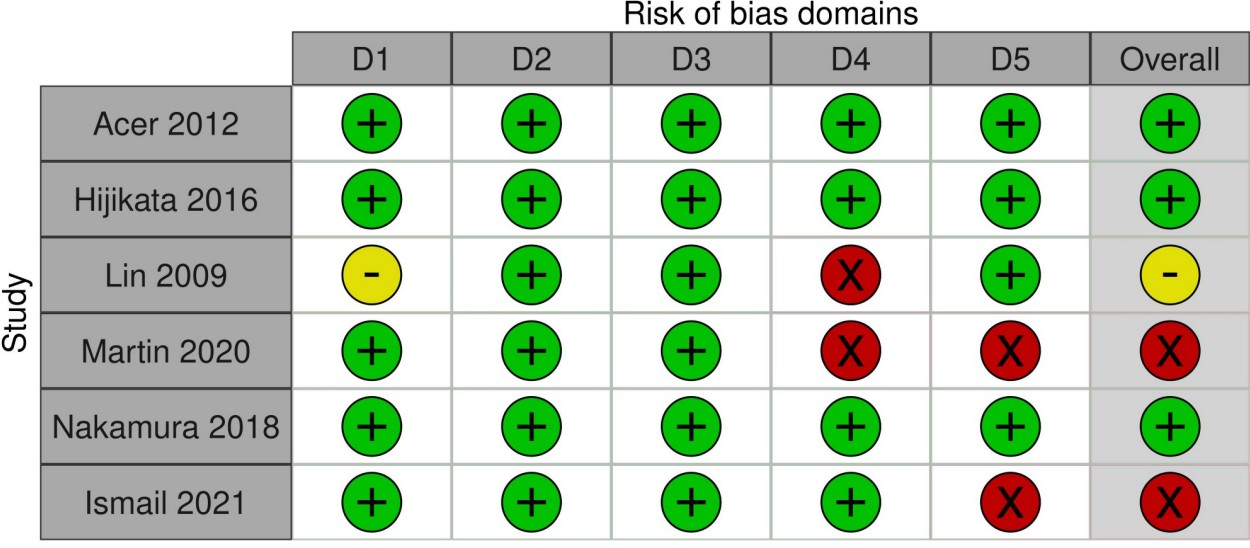

Domains:
D1: Bias arising from the randomization process.
D2: Bias due to deviations from intended intervention.
D3: Bias due to missing outcome data.
D4: Bias in measurement of the outcome.
D5: Bias in selection of the reported result.

Judgement
X High
- Some concerns
+ Low

**Fig 2. Risk of bias in included trials.**

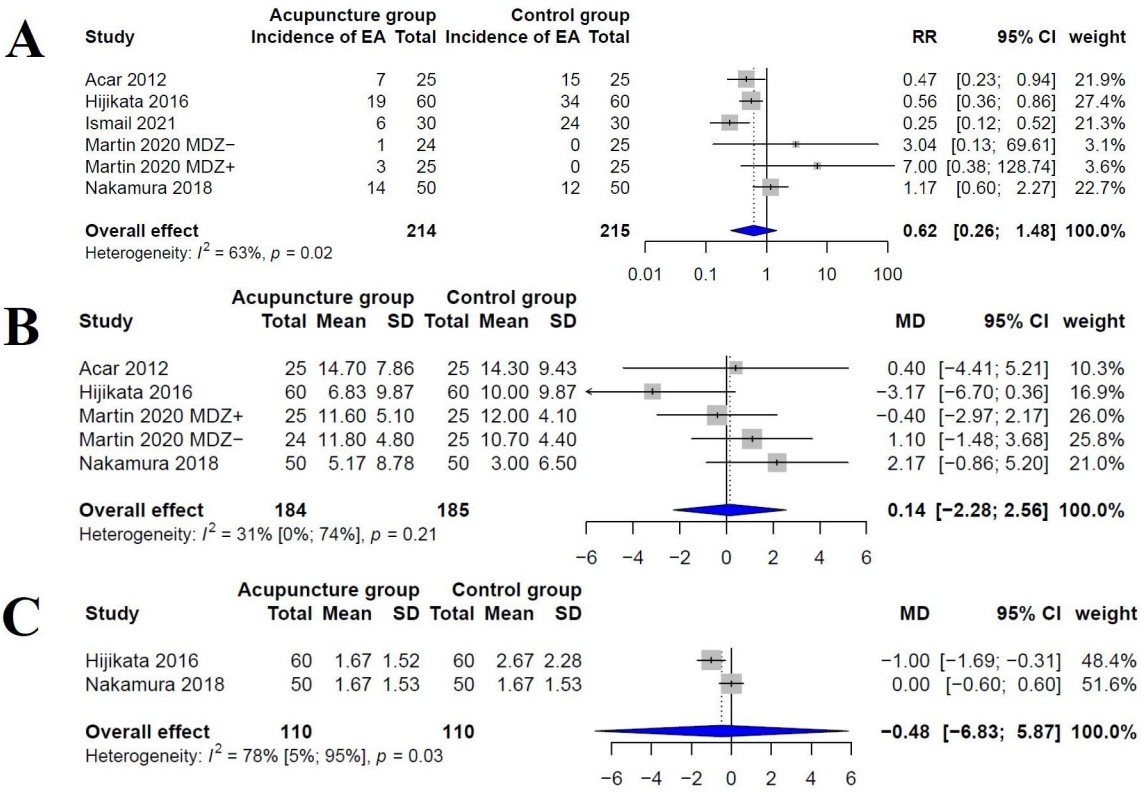

**Fig 3.** A. Forest plot showing incidence of anesthesia emergence (EA) using Pediatric Anesthesia Emergence Delirium (PAED) scale scores. B. Forest plot showing Pediatric Anesthesia Emergence Delirium (PAED) scale scores. C. Forest plot showing Aono's scale scores.

23.4%, respectively. The combined result showed no significant difference (RR, 0.62; 95% CI, 0.26–1.48; $I^2$ = 63%, Fig 3A). The absolute values of the PAED scale scores in the acupuncture therapy and control groups were 8.74 and 9.05, respectively. The combined result showed no significant difference (MD, 0.14; 95% CI, -2.28–2.56; $I^2$ = 31%, Fig 3B). Two trials evaluated the incidence of EA using Aono's scale [3, 12]. The absolute values of the Aono's scale scores in the acupuncture therapy and control groups were 1.67 and 2.22, respectively, which showed no significant difference (MD, -0.48; 95% CI, -6.83–5.87; $I^2$ = 78%, Fig 3C).

## Intervention effect for pain

Four trials evaluated pain using the Children's Hospital of Eastern Ontario Pain Scale (CHE-OPS) score [3, 10–12]. The absolute values of the CHEOPS scores in the acupuncture therapy and control groups were 8.35 and 8.73, respectively, which showed no significant difference (MD, -0.41; 95% CI, -4.27–3.46; $I^2$ = 66%, S1A Fig in S1 File).

## Adverse events

No increase in adverse events was reported in the acupuncture group. Additionally, no significant difference was observed between the two groups regarding the time taken for extubation (MD, -0.04; 95% CI, -11.92–11.18; $I^2$ = 12%, S1B Fig in S1 File) and PACU length of stay (MD, 1.76; 95% CI, -10.01–13.53; $I^2$ = 81%, S1C Fig in S1 File).

## Small-study effects

We could not conduct an asymmetry test for the funnel plot because only five trials were included.

## Sensitivity analysis

We conducted a sensitivity analysis according to the risk of bias. No statistically significant difference was observed when only trials with a low risk of bias were included (RR, 0.66; 95% CI, 0.21–2.08; $I^2$ = 54%, S2 Fig in S1 File).

## Subgroup analysis

We conducted a predefined subgroup analysis to explore the causes of heterogeneity, including the type of acupuncture therapy, selection of points (unilateral or bilateral), and surgery type (high-risk surgery for EA or not). The subgroup analysis for the type of surgery showed that the heterogeneity decreased in both groups, and the interaction of surgery type and acupuncture therapy was significant (interaction p < 0.01, S3 Fig in S1 File). All point estimates of the risk ratio were less than 1 in the subgroups for high-risk surgery; conversely, all point estimates of the risk ratio were greater than 1 in the subgroups for low-risk surgery. The 95% confidence intervals were wide for both subgroups (RR, 0.43; 95% CI, 0.16–1.18 in the High-risk group: RR, 1.32; 95% CI, 0.37–4.71 in the Low-risk group, S3 Fig in S1 File). In other subgroup analyses, heterogeneity remained, with no observed significant interaction (S4, S5 Figs in S1 File).

## Trial sequential analysis

Trial sequential analysis for the incidence of EA showed that the estimated required information size was 2,435; however, the collated information size was lower than this at 429 (17.6%). The Z curve did not cross the TSA monitoring boundary or reach the required information size (Fig 4). This indicates that the accumulated data were insufficient to conclusively determine the effectiveness of acupuncture therapy in treating EA in surgical pediatric patients.

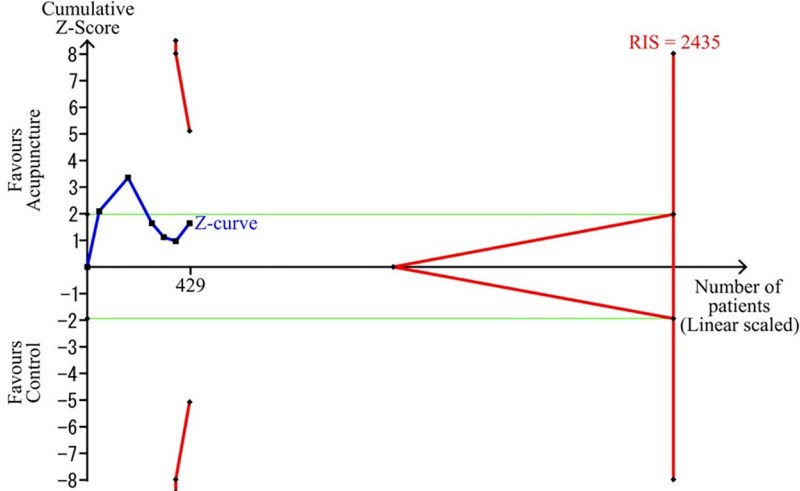

**Fig 4. Trial sequential analysis.**

## Quality of evidence

We evaluated the quality of evidence using the GRADE system. The evidence quality for the incidence of EA was "very low" as there were limitations in study design, inconsistency, imprecision, and possible publication bias. Indirectness was not detected (Table 2).

## Discussion

Our results showed that the current accumulated RCT data are insufficient to conclude the effectiveness of acupuncture therapy in preventing EA in children undergoing general anesthesia. The combined results showed that acupuncture therapy reduced the incidence of EA at a risk ratio of 0.62 with a wide confidence interval; additionally, the TSA indicated that further studies are needed to confirm this result. Consequently, the quality of the assessed evidence based on the GRADE approach was "very low." Additionally, the secondary outcomes, namely the absolute mean values of EA and pain scores evaluated using specific assessment tool scales, showed no significant differences. No trial reported adverse side effects or minor transient side effects associated with acupuncture therapy.

In 2009, Lin et al. [11] reported the first RCT regarding this topic; subsequently, four more RCTs have been published. All five trials included in this meta-analysis differed in terms of the method of acupuncture therapy, acupuncture point selection, and surgery type. Therefore, the integrated results showed moderate heterogeneity, with $I^2 > 50\%$. We conducted a subgroup analysis to clarify the cause of heterogeneity, which showed that acupuncture therapy was associated with the type of surgery; particularly, it tended to prevent EA in high-risk surgeries for EA. The interaction p-value was significant, suggesting the presence of a qualitative interaction between the two effects of acupoint stimulation. However, this result has wide 95% confidence

**Table 2. Summary of evidence.**

**Acupuncture therapy compared to placebo/sham or standard care for preventing EA**

**Patient or population**: preventing EA

**Setting**: Operation room

**Intervention**: acupuncture therapy

**Comparison**: placebo/sham or standard care

| Outcome № of participants (studies) | Relative effect (95% CI) | Anticipated absolute effects (95% CI) | | | Certainty |
|---|---|---|---|---|---|
| | | | | Difference | |
| Incidence of EA № of participants: 429 (5 RCTs) | **OR 0.62** (0.26 to 1.48) | 39.5% | **23.4%** (11.1 to 42.1) | **16.1% fewer** (28.5 fewer to 2.5 more) | ⊕○○○ VERY LOW [a,b,c,d] |

**The risk in the intervention group** (and its 95% confidence interval) is based on the assumed risk in the comparison group and the **relative effect** of the intervention (and its 95% CI).

**Grading of Recommendations Assessment, Development, and Evaluation (GRADE) Working Group grades of evidence**

**High certainty**: We are very confident that the true effect lies close to that of the estimate of the effect.

**Moderate certainty**: We are moderately confident in the effect estimate: The true effect is likely to be close to the estimate of the effect, but there is a possibility that it is substantially different.

**Low certainty**: Our confidence in the effect estimate is limited: The true effect may be substantially different from the estimate of the effect.

**Very low certainty**: We have very little confidence in the effect estimate: The true effect is likely to be substantially different from the estimate of effect.

[a]. Only three trials were at low risk of bias.

[b]. There is a high clinical heterogeneity.

[c]. The trial sequential analysis (TSA)-adjusted CI was wide.

[d]. Publication bias could not be assessed because only five trials were included.

**EA**: Emergence agitation; **CI**: Confidence interval; **RR**: Risk ratio; **OR**: Odds ratio; **MD**: Mean difference

intervals and is not conclusive; therefore, it should be regarded as a hypothesis-generating result. Future studies limited to surgeries with a high risk for EA should be conducted to confirm this hypothesis.

We considered that the preventive effect of acupuncture on EA might be associated with pain reduction; therefore, we analyzed the CHEOPS score outcomes as secondary endpoints. A previous study found a strong relationship between EA and postoperative pain [29]. In our study, no significant differences were found in any of the secondary outcomes and adverse events. However, the included trials showed significant differences regarding surgery type and postoperative pain management methods; therefore, further trials are needed to evaluate the effect of acupuncture therapy on pain.

Additionally, we assessed the incidence of adverse events such as nausea and vomiting, time taken for extubation, and PACU length of stay. There was no increase in the adverse events reported in the included studies in the acupuncture therapy group, with no significant difference between groups. The National Institute of Health Consensus states that under similar conditions, acupuncture therapy causes substantially lower adverse effects compared to other drugs and other accepted procedures [13], which is consistent with our results. Our findings suggested that the side effects of acupuncture therapy are minimal; however, the results, based only on RCTs, are insufficient to evaluate rare adverse events. Although adverse events are expected to be minimal, the clinical application of acupuncture for EA prevention purposes should be withheld until efficacy is firmly confirmed in future RCTs and meta-analyses.

Our study has several limitations. First, only six trials were included in the meta-analysis. The total number of patients was considerably lower compared to the RIS, and the possibility of publication bias could not be ruled out. The results showed moderate heterogeneity ($I^2 >$ 50%), and we found that one out of four of these trials had a high risk of bias. Thus, we downgraded the quality of evidence to "very low." Second, there is high clinical heterogeneity due to differences in acupuncture therapy, point selection, and surgery type. The integration of results from trials using different methodologies and subjects remains controversial. $I^2$, an index of heterogeneity, was high; however, subgroup analysis suggested that some of the heterogeneity was created by the difference in the surgery type (high or low-risk surgery). This finding might generate an attractive hypothesis that the effect of acupuncture therapy may vary depending on the surgery. Third, all studies in this research chose to stimulate HT7 as the acupoint. While this may have reduced clinical heterogeneity in terms of treatment, it is important to note that our study results cannot be extrapolated to the stimulation of other acupoints.

## Conclusion

In conclusion, this meta-analysis showed that the currently available RCTs are insufficient to conclusively determine the effectiveness of acupuncture therapy, especially HT7 stimulation, in preventing EA in children undergoing general anesthesia. The quality of evidence from previous RCTs was "very low."

## Supporting information

**S1 Checklist. PRISMA checklist.**
(DOCX)

**S1 File.**
(DOCX)

## Acknowledgments

We would like to thank Editage (www.editage.com) for English language editing.

## Author Contributions

**Conceptualization:** Takahiro Mihara.

**Data curation:** Daisuke Nakajima, Toshiyuki Hijikata.

**Formal analysis:** Takahiro Mihara, Daisuke Nakajima.

**Investigation:** Takahiro Mihara.

**Methodology:** Takahiro Mihara, Daisuke Nakajima, Makoto Tomita.

**Project administration:** Takahiro Mihara.

**Validation:** Daisuke Nakajima.

**Visualization:** Daisuke Nakajima.

**Writing – original draft:** Takahiro Mihara, Daisuke Nakajima.

**Writing – review & editing:** Takahiro Mihara, Toshiyuki Hijikata, Makoto Tomita, Takahisa Goto.

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
