## [Decision Letter · Decision Letter 0]

20 Feb 2023

PONE-D-22-28712Effectiveness of acupuncture therapy for the prevention of emergence agitation in children: A systematic review and meta-analysis with trial sequential analysisPLOS ONE

Dear Dr. Mihara,

Thank you for submitting your manuscript to PLOS ONE. After careful consideration, we feel that it has merit but does not fully meet PLOS ONE’s publication criteria as it currently stands. Therefore, we invite you to submit a revised version of the manuscript that addresses the points raised during the review process.

We look forward to receiving your revised manuscript.

Kind regards,

Huijuan Cao, Ph.D.

Academic Editor

PLOS ONE

Journal Requirements:

Additional Editor Comments:

Some details were not provided, e.g. the inclusion criteria of the review. The primary outcome was incidence of EA, however, the diagnostic criteria of EA was not specified. It's very important to identify how to determine EA accroding to the PACU scores. Besides, in Table 1 the type of control in each included trial should also be reported.

Purpose of this review is to assess the  the effectiveness of acupuncture therapy in preventing EA in children undergoing general anesthesia, and the current results showed no difference between groups. With limited number of included participants, results of subgroup analysis may not really suggest the conclusions in the text. I suggest that authors carefully evaluate the existing evidence and draw objective conclusions.

Reviewers' comments:

Reviewer's Responses to Questions

**Comments to the Author**

1. Is the manuscript technically sound, and do the data support the conclusions?

Reviewer #1: Yes

Reviewer #2: Yes

2. Has the statistical analysis been performed appropriately and rigorously? 

Reviewer #1: Yes

Reviewer #2: Yes

3. Have the authors made all data underlying the findings in their manuscript fully available?

Reviewer #1: Yes

Reviewer #2: Yes

4. Is the manuscript presented in an intelligible fashion and written in standard English?

Reviewer #1: Yes

Reviewer #2: Yes

5. Review Comments to the Author

Reviewer #1: I do not have any concerns about this paper and all of the criteria for publication have been met. This paper appropriately identifies the lack of evidence for and high quality studies of acupuncture for EA.

Reviewer #2: This study is an important report for improving the quality of integrative medicine in recent years. In particular, acupuncture therapy is noninvasive and is expected to play a role as an adjunctive treatment. Thus, it is necessary to accumulate scientific evidence and information, and the authors have organized the data for acupuncture therapy in preventing emergence agitation. Although the data appear to be factual, there are a few questions.

The selected papers seem to be a collection of reports that chose the acupuncture point of HT7 (Shenmen). Why is HT7 regularly used in previous studies? Please describe in Background or Discussion.

Some are written in the limits (L300), acupuncture treatment focused on HT7 is just ineffective, other acupuncture points may be effective? Can you suggest other acupoints that are more effective than HT7 from an oriental medicine side?

Shouldn't the Conclusion be "acupuncture treatment focusing on HT7" instead of "effects of acupuncture"? Please reconsider your conclusions along with the contents of the excluded papers.

This research and analysis will hopefully lead to a stronger and better culture of acupuncture.

6. PLOS authors have the option to publish the peer review history of their article (what does this mean?). If published, this will include your full peer review and any attached files.

Reviewer #1: No

Reviewer #2: No

---

## [Author Response · Author response to Decision Letter 0]

22 Mar 2023

Our responses to each of your comments are provided in a separate file, titled "Response to Reviewers."

---

## [Decision Letter · Decision Letter 1]

7 May 2023

PONE-D-22-28712R1Effectiveness of acupuncture therapy for the prevention of emergence agitation in children: A systematic review and meta-analysis with trial sequential analysisPLOS ONE

Dear Dr. Mihara,

Thank you for submitting your manuscript to PLOS ONE. After careful consideration, we feel that it has merit but does not fully meet PLOS ONE’s publication criteria as it currently stands. Therefore, we invite you to submit a revised version of the manuscript that addresses the points raised during the review process.

We look forward to receiving your revised manuscript.

Kind regards,

Huijuan Cao, Ph.D.

Academic Editor

PLOS ONE

Journal Requirements:

Reviewers' comments:

Reviewer's Responses to Questions

**Comments to the Author**

1. If the authors have adequately addressed your comments raised in a previous round of review and you feel that this manuscript is now acceptable for publication, you may indicate that here to bypass the “Comments to the Author” section, enter your conflict of interest statement in the “Confidential to Editor” section, and submit your "Accept" recommendation.

Reviewer #2: All comments have been addressed

Reviewer #3: All comments have been addressed

2. Is the manuscript technically sound, and do the data support the conclusions?

Reviewer #2: Yes

Reviewer #3: Yes

3. Has the statistical analysis been performed appropriately and rigorously? 

Reviewer #2: Yes

Reviewer #3: Yes

4. Have the authors made all data underlying the findings in their manuscript fully available?

Reviewer #2: No

Reviewer #3: Yes

5. Is the manuscript presented in an intelligible fashion and written in standard English?

Reviewer #2: Yes

Reviewer #3: Yes

6. Review Comments to the Author

Reviewer #2: (No Response)

Reviewer #3: The points highlighted in previous review have been successfully addressed. However, I would like to raise one concern to improve the quality of this manuscript. please describe the ‘emergence agitation’ in more details in the introduction section.

7. PLOS authors have the option to publish the peer review history of their article (what does this mean?). If published, this will include your full peer review and any attached files.

Reviewer #2: No

Reviewer #3: No

---

## [Author Response · Author response to Decision Letter 1]

8 May 2023

Our responses to each of reviewer's comments are provided in a separate file, titled "Response to Reviewers."

---

## [Editor Report · Decision Letter 2]

24 May 2023

Effectiveness of acupuncture therapy for the prevention of emergence agitation in children: A systematic review and meta-analysis with trial sequential analysis

PONE-D-22-28712R2

Dear Dr. Takahiro Mihara,

We’re pleased to inform you that your manuscript has been judged scientifically suitable for publication and will be formally accepted for publication once it meets all outstanding technical requirements.

Kind regards,

Huijuan Cao, Ph.D.

Academic Editor

PLOS ONE
---

## [Editor Report · Acceptance letter]

29 May 2023

PONE-D-22-28712R2 

Effectiveness of acupuncture therapy for the prevention of emergence agitation in children: A systematic review and meta-analysis with trial sequential analysis 

Dear Dr. Mihara:

I'm pleased to inform you that your manuscript has been deemed suitable for publication in PLOS ONE. Congratulations! Your manuscript is now with our production department. 

Kind regards, 

on behalf of

Dr. Huijuan Cao 

Academic Editor

PLOS ONE